# Review of Peste des Petits Ruminants Occurrence and Spread in Tanzania

**DOI:** 10.3390/ani11061698

**Published:** 2021-06-07

**Authors:** Daniel Pius Mdetele, Erick Komba, Misago Dimson Seth, Gerald Misinzo, Richard Kock, Bryony Anne Jones

**Affiliations:** 1Department of Veterinary Services, Ministry of Livestock and Fisheries, P.O. Box 2870, 25526 Dodoma, Tanzania; 2Department of Veterinary Medicine and Public Health, College of Veterinary Medicine and Biomedical Sciences, Sokoine University of Agriculture, P.O. Box 3021, Chuo Kikuu, 25523 Morogoro, Tanzania; komba.erick@gmail.com; 3National Institute for Medical Research, Tanga Medical Research Centre, 25527 Tanga, Tanzania; sethmdj@gmail.com; 4Department of Veterinary Microbiology, Parasitology and Biotechnology, College of Veterinary Medicine and Biomedical Sciences, Sokoine University of Agriculture, P.O. Box 3019, Chuo Kikuu, 25523 Morogoro, Tanzania; gerald.misinzo@sacids.org; 5Royal Veterinary College, Hawkshead Lane, Hatfield, Herts AL9 7TA, UK; rkock@rvc.ac.uk (R.K.); bryony.jones@apha.gov.uk (B.A.J.)

**Keywords:** Peste des petits ruminants, Peste des petits ruminants virus, transboundary animal diseases, epidemiology, surveillance, sheep, goat, small ruminant

## Abstract

**Simple Summary:**

Peste des petits ruminants (PPR), caused by PPR virus (PPRV), is a transboundary animal disease of sheep and goats that has a significant impact on farmer’s livelihoods, food and nutritional security; and threatens susceptible wildlife. This review compiled information on the introduction and spread of PPR in Tanzania, from published and unpublished sources. PPR was first confirmed in Tanzania in 2008, but could have been present earlier, based on antibody detection in archived sera. The virus was probably introduced to northern Tanzania through cross-border movement of sheep and goats, and afterwards spread to eastern, central and southern Tanzania through movement of animals by pastoralists and traders. Genome sequencing shows that there have been several introductions of PPRV and it is now considered to be endemic. PPR has not been observed in cattle, camels or wildlife, but sera collected from these species contain PPRV antibodies, indicating virus exposure, probably through contact with infected sheep and goats. Some challenges for PPR control in Tanzania include the spread of the disease through small ruminants movements for pastoralism and trade, and limited veterinary services for disease surveillance and vaccination. The socio-economic impact of PPR justifies investment in a comprehensive disease eradication programme.

**Abstract:**

Peste des petits ruminants (PPR) is an important transboundary animal disease of domestic small ruminants, camels, and wild artiodactyls. The disease has significant socio-economic impact on communities that depend on livestock for their livelihood and is a threat to endangered susceptible wild species. The aim of this review was to describe the introduction of PPR to Tanzania and its subsequent spread to different parts of the country. On-line databases were searched for peer-reviewed and grey literature, formal and informal reports were obtained from Tanzanian Zonal Veterinary Investigation Centres and Laboratories, and Veterinary Officers involved with PPR surveillance were contacted. PPR virus (PPRV) was confirmed in northern Tanzania in 2008, although serological data from samples collected in the region in 1998 and 2004, and evidence that the virus was already circulating in Uganda in 2003, suggests that PPRV might have been present earlier than this. It is likely that the virus which became established in Tanzania was introduced from Kenya between 2006–7 through the cross-border movement of small ruminants for trade or grazing resources, and then spread to eastern, central, and southern Tanzania from 2008 to 2010 through movement of small ruminants by pastoralists and traders. There was no evidence of PPRV sero-conversion in wildlife based on sera collected up to 2012, suggesting that they did not play a vectoring or bridging role in the establishment of PPRV in Tanzania. PPRV lineages II, III and IV have been detected, indicating that there have been several virus introductions. PPRV is now considered to be endemic in sheep and goats in Tanzania, but there has been no evidence of PPR clinical disease in wildlife species in Tanzania, although serum samples collected in 2014 from several wild ruminant species were PPRV sero-positive. Similarly, no PPR disease has been observed in cattle and camels. In these atypical hosts, serological evidence indicates exposure to PPRV infection, most likely through spillover from infected sheep and goats. Some of the challenges for PPRV eradication in Tanzania include movements of small ruminants, including transboundary movements, and the capacity of veterinary services for disease surveillance and vaccination. Using wildlife and atypical domestic hosts for PPR surveillance is a useful indicator of endemism and the ongoing circulation of PPRV in livestock, especially during the implementation of vaccination to control or eliminate the disease in sheep and goats. PPR disease has a major socio-economic impact in Tanzania, which justifies the investment in a comprehensive PPRV eradication programme.

## 1. Introduction

Peste des petits ruminants (PPR) is a highly contagious and economically important viral disease of domestic small ruminants [1]. It can also cause severe disease and mortality in some wild artiodactyl species and is a threat to biodiversity conservation [2,3,4], and can infect other atypical domestic species such as cattle, camels and pigs. It is caused by *Small ruminant morbillivirus* (commonly known as PPR virus) of the genus *Morbillivirus* and the family *Paramyxoviridae*, and has been classified into four genetically distinct lineages (I, II, III and IV) based on a partial sequence analysis of the fusion protein (F) and nucleoprotein (N) genes [5,6]. PPR virus (PPRV) was first identified in Côte d’Ivoire, West Africa, in 1942, and it is currently believed to be endemic across much of West, Central, North and East Africa, the Middle East and Central, South and East Asia [7]. Geographically, lineages I and II have been found predominantly in West and Central Africa, and lineage III has been found predominantly in East Africa and the Middle East [8]. Lineage IV is the main lineage found in Asia [9], both in wild and domestic small ruminants [10,11,12], and more recently it has been found in Africa [5,13,14].

Clinically, the disease in sheep and goats is characterised by a high fever, catarrhal ocular discharges, mucopurulent nasal discharges and erosive stomatitis in the early stages, followed by severe enteritis and bronchopneumonia [15]. The morbidity and mortality rates in PPRV endemic areas are lower compared to the rates observed in PPRV epidemics in non-endemic areas [1]. Mortality is higher in young animals compared to adults in both endemic and epidemic settings, and goats tend to be more severely affected than sheep [15,16]. Similar clinical signs have been observed in PPR outbreaks affecting various wild ruminant species in Asia, in captive, managed free-range, and wild living populations [2,4]. PPRV appears to spillover from infected domestic sheep and goats into wild animals that are in proximity, leading to infection and, in some cases, clinical disease [3,4,10,17]. The virus also causes sub-clinical infection in cattle, which develop antibodies against PPRV [18,19].

PPRV is transmitted primarily through direct contact between infected and susceptible animals, therefore communal grazing areas and live animal markets are important places for the spread of the virus [8,20]. Large amounts of infective virus are excreted in secretions and discharges from the eyes, nose and mouth, as well as in faeces [16,21,22], and the primary route of infection is respiratory, by short-range aerosols generated by sneezing and coughing [23]. The virus is fragile, so transmission of the virus by fomites is unlikely [23]. Animals that are infected with PPRV start to develop antibodies from seven to ten days post-infection [16]. Those animals that recover from infection develop lifelong immunity that is fully protective against reinfection [23], and vaccination with live attenuated PPRV vaccine also provides lifelong protection [24,25]. The offspring of immune animals have protective maternal antibodies for up to three to four months of age [26,27,28].

Africa and Asia have approximately 80% of the global sheep and goat population [29]. In low and middle-income countries, goats and sheep are an important resource for small-holder farmers and pastoralists, especially in arid and semi-arid areas because of their low cost, and their ability to survive on sparse pastures and withstand drought [30].

In eastern Africa, PPR disease was first reported in Sudan in 1972 [31] and then in the horn of Africa, Ethiopia in 1994 [32]. In East Africa, Kenya officially confirmed the occurrence of PPR disease in 2007, although it is possible that disease incursions in northern districts occurred during the 1990s based on clinical and serological evidence [33,34]. This was a time of active rinderpest circulation [35] and PPRV serology may have been unreliable with cross-reactivity between these closely related viruses in the absence of attempts for differentiation by cross-neutralisation [34]. Uganda confirmed PPR disease officially for the first time in 2007 [36,37], but evidence for infection in Uganda was first reported in small ruminants in 2003 and in wildlife based on serology in 2004 [38]; while Tanzania first confirmed PPR disease the following year, in 2008 [39]. The first confirmation of PPR disease in the Democratic Republic of the Congo (DRC) was in 2012, and in Burundi in 2017 [40]. However, there have so far been no confirmed cases of PPR disease in Rwanda [41], but serological evidence suggests PPRV could be present although it is still unconfirmed by virus detection [42]. No clinical disease has been reported in Malawi, Mozambique or Zambia, which border Tanzania to the south [43].

The Global Strategy for the Control and Eradication of PPR (PPR GCES) was officially adopted in 2015 by the World Organization for Animal Health (OIE) and the Food and Agriculture Organization of the United Nations (FAO). The strategy describes the rationale for controlling and eradicating PPRV, the general principles, and the tools to be used [1]. A strategy for Africa has been developed by the African Union Inter-African Bureau for Animal Resources (AU-IBAR), and regional strategies that are aligned with the global strategy have been developed by the Inter-governmental Authority on Development (IGAD) for the Greater Horn region, and the Southern African Development Community (SADC) [43,44,45]. At the national level, Tanzania is in the process of developing a national PPRV control and eradication plan.

The aim of this review is to describe the first confirmed introduction of PPRV to Tanzania and its subsequent persistence and spread to different parts of the country. This review will contribute to an improved understanding of the history of PPRV in Tanzania and will provide insights into how the virus is maintained and spread, that will contribute to more effective planning and implementation of the PPRV control and eradication programme by the Tanzania Veterinary Services and the veterinary services of neighbouring countries.

## 2. Materials and Methods

A narrative review method was used, and a literature search for peer-reviewed published papers and grey literature was carried out for all documentation up to 2020 using the online publication databases; CAB Direct (https://www.cabdirect.org/ accessed on 2 March 2020), PubMed (https://www.ncbi.nlm.nih.gov/pubmed/ accessed on 20 March 2020) and Google Scholar (https://scholar.google.com/ accessed on 20 March 2020). The search strategy focused on reports of PPR disease surveillance, investigations in Tanzania, and the location of PPR disease events, prior to and after the first confirmation of PPRV in 2008, using the search terms “peste des petits ruminants” and “Tanzania”. For the published articles, only articles written in English were considered and 19 publicly available papers were included in this review. For the unpublished reports, documents in English and Swahili languages were considered, and 10 reports were included in this review. The titles of papers and documents generated by the searches were reviewed and the full texts of those that were relevant were obtained for full text review. If the title did not provide enough detail for acceptance or rejection, the abstract was also scrutinised. To be considered for review, the papers had to provide information on the occurrence of PPRV infection and/or disease in Tanzania. The first author, along with two other co-authors selected the articles to be reviewed, and the remaining authors were involved in providing additional literature.

In addition to publicly available literature, the annual reports (2008–2018) of the seven Zonal Veterinary Investigation Centres (VIC) were reviewed for information about PPRV, and formal and informal reports, letters and emails containing information about PPR surveillance or disease events between 2007 and 2018 were obtained from the files of the three Zonal VICs and zonal veterinary laboratories that are responsible for disease surveillance in North and Central Tanzania (Northern, Lake and Central Zones) (Figure 1). Veterinary officers at the Northern, Lake and Central Zonal VICs and Laboratories, who had been involved in PPR surveillance and investigations, were contacted by phone, or were met in person to obtain information about current or unpublished studies.

## 3. Results and Discussion

The results of this review are presented as follows. First, we describe the evidence available for the possible presence of PPRV prior to official confirmation in 2008. Then we describe the PPR disease outbreak that occurred in 2008 in northern Tanzania, followed by the emergence of PPRV disease in southern Tanzania between 2009 and 2010, in eastern Tanzania in 2010, and in Central Tanzania in 2014. This is followed by descriptions of the evidence of PPRV infection in other domestic animals and in wild species, the molecular characterization of PPRV obtained from cases in different parts of the country, PPRV serological surveys, and evidence of PPRV co-infection with other pathogens in Tanzania. Finally, we describe the socio-economic impact and the main challenges for PPRV control in Tanzania.

### 3.1. Evidence of PPRV in Tanzania Prior to Confirmation of Infection in 2008

Wambura (2000) reported that, prior to 1998, no PPR outbreaks had been reported, and no serological surveys had been carried out in Tanzania [46]. In 1998, a national cross-sectional serological survey for PPRV and rinderpest virus was conducted. A total of 3134 serum samples were collected from sheep and goats that were randomly selected from all 20 regions in all seven surveillance zones of mainland Tanzania (Figure 1). The samples were analysed for both rinderpest virus and PPRV antibodies using the rinderpest virus haemagglutinin (H) competitive enzyme-linked immunosorbent assay (c-ELISA) and the PPRV H c-ELISA [47], and all were negative in both tests, including 520 samples from Arusha region where the first confirmed outbreak of PPRV was subsequently detected in 2008 [46]. Rinderpest virus is a morbillivirus that is closely related to PPRV, for which cattle were the main host, but sheep and goats could be infected with rinderpest virus and develop antibodies. The global eradication of rinderpest was achieved in 2011. PPRV antibodies could cross-react with rinderpest virus c-ELISA, therefore it was necessary to run both assays in parallel to distinguish between PPRV and rinderpest virus antibodies.

Serological surveillance was conducted in wildlife populations across East and Central Africa between 1994 and 2004 during the rinderpest eradication programmes of the Pan-African Rinderpest Campaign (PARC) and the Pan-African Programme for the Control of Epizootics (PACE) [48]. While the main objective was surveillance for rinderpest virus antibodies, 967 serum samples from 20 wild artiodactyl species were also tested for PPRV antibodies on plates coated with a PPRV recombinant nucleocapsid protein (N) c-ELISA at the Agricultural Research Centre for International Development (CIRAD), France. A subset was also tested by PPRV H c-ELISA at The Pirbright Institute, UK. A rinderpest virus and PPRV cross-neutralisation test was carried out for PPRV N c-ELISA positive samples, and 31 were considered to be PPRV sero-positive based on higher titres in the PPR virus neutralization test (VNT) compared to rinderpest VNT. These positive samples were collected from African buffalo (*Syncerus caffer*) and eland (*Taurotragus oryx*) in southern Ethiopia and Kenya. Out of the 967 samples, 127 were collected in Tanzania from buffalo (92), eland (15), giraffe (*Giraffa camelopardalis*) (9), hartebeest (*Alcelaphus buselaphus*) (4), kudu (*Tragelaphus strepsiceros*) (2), oryx (*Oryx gazella*) (1), roan antelope (*Hippotragus equinus*) (2), sable antelope (*Hippotragus niger*) (1) and topi (*Damaliscus lunatus jimela*) (1) in the Serengeti and Mkomazi protected areas in northern Tanzania, and in the Ruaha and Katavi protected areas in southern Tanzania. All of these samples were PPRV sero-negative by N c-ELISA and one buffalo and two eland from Mkomazi and one buffalo from Serengeti were PPRV sero-positive by H cELISA but also positive for rinderpest by H c-ELISA and serum neutralization, so these results were considered to be due to cross-reaction [48].

Karimuribo et al. (2010) carried out a retrospective investigation to determine whether PPRV might have been present in northern Tanzania prior to the official confirmation in 2008 [49]. A total of 198 serum samples were retrieved from the serum bank of the VIC in Arusha and were analysed for PPRV antibodies by N c-ELISA as described by Swai et al. (2009) [50]. These serum samples had been collected from goats and sheep during investigations in the Ngorongoro district of suspected Rift Valley fever (RVF) cases in 1998 (52 serum samples) and suspected PPR cases in 2004 (21 samples), and during a toxoplasmosis survey in 2004 covering the Ngorongoro, Monduli, Mbulu and Karatu districts (125 samples). All the samples collected in 1998 were sero-negative, but 17.1% of sera collected in 2004 were sero-positive (25 of 146 samples). There was a higher sero-prevalence (71.4% of 21 samples) for the serum samples that were collected from suspected PPR cases compared to those collected for investigations of other diseases (5.7%, χ^2^ test *p* value < 0.001). Moreover, the animals sampled in the Ngorongoro district had a higher sero-prevalence (18.3% of 104 samples) compared to the animals sampled from other areas (6.4% of 94 samples). Interviews were conducted with key informants from the Ngorongoro district veterinary office, VIC Arusha and the epidemiology unit in the Ministry of Livestock Development and Fisheries, who provided information on reports and investigations of PPR-like diseases in Ngorongoro district in 1995, 2002, 2004 (when the above serum samples were collected), 2006 and 2008 (when PPRV was confirmed). The authors concluded that PPRV had been present in northern Tanzania at least four years prior to the official confirmation in 2008 [49]. An alternative explanation was cross-reactivity to rinderpest virus in small ruminants. Rinderpest disease in cattle was confirmed in Karatu district, and in Ngorongoro Conservation Area and Loliondo in Ngorongoro district in 1997, and sero-surveillance showed a low sero-prevalence of rinderpest antibody in small ruminants at that time [51]. Spillover of rinderpest virus to small ruminants could account for cross-reactive PPRV sero-positivity and for the PPR-like clinical signs observed, when rinderpest virus was present and actively circulating in cattle in northern Tanzania and across the border in Kenya. If the animals that were sero-positive in 2004 were alive at the time of rinderpest circulation, it is possible that the positive results were a cross-reaction due to the presence of rinderpest virus antibody.

### 3.2. The First Confirmed PPRV Disease Cases in Northern Tanzania, 2008

Various sources describe reports of a PPR-like disease occurring in Ngorongoro district in Arusha Region of northern Tanzania in late 2007 and early 2008 [39,49,52]. In March 2008, high mortality was reported among sheep and goats in Ngorongoro district, leading to an investigation by VIC Arusha during which a total of 112 sheep and goats were clinically examined, some post mortem examinations were carried out, and serum samples were collected and found to be antibody negative to PPRV [39]. Unfortunately, serological analysis at the Central Veterinary Laboratory (CVL) was delayed for approximately six months due to a lack of diagnostic test kits [52]. Reports of sheep and goat mortality continued in Ngorongoro and Mara districts, so in June 2008 a further investigation was carried out. A total of 404 serum samples were collected in Ngorongoro district, of which 129 (31.9%) were sero-positive, and 84 sera were collected in Mara district, of which all were sero-negative [39].

The serological results suggested the presence of PPRV in northern Tanzania, therefore a cross-sectional serological survey was carried out between August 2008 and July 2009 in 12 purposively selected districts of northern Tanzania that were perceived to be at risk of PPRV due to their proximity to southern Kenya, where PPR disease had recently been reported in Kajiado district. The aim of the survey was to investigate whether there was evidence of PPRV circulation and the spatial extent of the spread [20,50]. A total of 3478 serum samples were collected from healthy unvaccinated goats (2182) and sheep (1296) from pastoralist or agropastoralist flocks in 79 randomly selected villages from four districts in Arusha region (Ngorongoro, Karatu, Longido, Monduli), one district in Kilimanjaro region (Siha), two districts in Manyara region (Mbulu, Simanjiro), and five districts of Tanga region (Korogwe, Lushoto, Mkinga, Muheza and Tanga). During the survey, cases of suspected PPR disease were observed in Ngorongoro, Monduli, Longido and Siha districts, with signs of diarrhoea, fever, and nasal discharge, and a post mortem examination of two goats found signs of bronchopneumonia, swollen mesenteric lymph nodes and hyperaemic small intestines [39]. Using the H c-ELISA [47], the overall sero-prevalence was 22.1% (95% Confidence Interval: 20.7–23.5%). The district level sero-prevalence varied from 0 to 88%, with a higher sero-prevalence in pastoral compared to agro-pastoral districts (Figure 2). In Arusha region, Longido district had the highest seroprevalence (88%) followed by Ngorongoro (55%), Monduli (35%) and Karatu districts (29%). In Manyara region, Mbulu district had a sero-prevalence of 47%, but Simanjiro was sero-negative. Siha district in Kilimanjaro region had a sero-prevalence of 43%. The district-level seroprevalence in Tanga Region was lower, ranging from 0 to 11% [39]. The high PPRV seroprevalence in an unvaccinated population suggested natural transmission of PPRV in these northern districts [50].

In December 2008, tissue samples including spleen, liver, mediastinal lymph nodes, and whole blood were collected from an outbreak in Soitsambu, Ngorongoro district, and sent to the World Reference Laboratory for PPR at CIRAD, France. PPRV was confirmed by reverse transcription polymerase chain reaction (RT-PCR), using primers targeting the nucleoprotein (N) gene, showing that this virus belonged to lineage III [39,52]. The first occurrence of PPRV in Tanzania was officially reported to the OIE on 27th January 2009 (https://wahis.oie.int/ accessed on 14 January 2020). The source of the virus was suspected to be through the cross-border movement of animals from Kenya, where PPRV had been confirmed in the border districts of Narok and Kajiado in 2008 [50,52,53].

During the same time period, VIC Mwanza received reports of suspected PPR outbreaks in the Lake Zone and alerted the District Veterinary Officers (DVOs) of all the districts in the zone, in particular those bordering the Northern Zone (Meatu, Bariadi, Serengeti, Bunda and Magu), bordering Kenya (Tarime and Rorya), and bordering Uganda (Misenyi and Karagwe). In December 2008, purposive serological surveys were conducted in Serengeti and Tarime districts of Mara region, and Bariadi and Meatu districts of Shinyanga region. In each of these districts, 10 villages were selected, and in each village, five livestock keepers were visited, and blood samples collected from 10 goats and sheep. A total of 1945 serum samples were collected and analysed by c-ELISA (type of cELISA not specified), out of which 189 (9.7%) were sero-positive. The proportion of animals that were sero-positive was low in animals sampled in Serengeti (0.8%), Tarime (3.1%) and Meatu districts (1.2%), but 33.7% of the animals sampled in Bariadi district were sero-positive, which suggests that PPRV had been circulating in this district, since no PPRV vaccination had been carried out in this area [20]. Bariadi district lies to the west of Ngorongoro district in the Northern Zone.

In response to the confirmed outbreaks of PPR in 2008, the Veterinary Services of Tanzania collaborated with the non-governmental organization (NGO), VETAID to carry out a vaccination campaign in February 2009 in Ngorongoro, Karatu, Mbulu, Monduli, Arumeru and Arusha districts of Arusha Region (62,000 doses). In the following year (January to April 2010), 3.2 million doses of vaccine were delivered to goats and sheep in the remaining districts of northern Tanzania by the Tanzania Veterinary Services with the assistance of FAO [52]. Another vaccination campaign was carried out from mid-2010 to 2012 by the Tanzania Veterinary Services with the support of the Vaccination for Control of Neglected Animal Diseases in Africa (VACNADA) project (a European Commission funded project that was coordinated by the African Union Interafrican Bureau for Animal Resources). Vaccinations were carried out in the regions neighbouring Kenya, covering districts in the Northern Zone (Kiteto, Simanjiro, Longido, Meru, Arusha DC, Monduli, Ngorongoro and Siha) and the Lake Zone (Tarime, Rorya, Bunda and Serengeti) [54] (Figure 2).

Since 2009, there have been several published reports of confirmed PPR disease in the Ngorongoro district [17,55,56,57]. In addition, unpublished CVL reports obtained from VIC Arusha provided information on confirmed PPR disease in Ngorongoro in 2017, and in the Kiteto district, Manyara region in 2015.

In August 2011, Mwanza VIC received a report of an outbreak of disease in sheep and goats causing mortality in villages around Lake Kitangiri in Meatu and Kishapu districts of Shinyanga region (Qwari, B., 2011, “Goat and sheep mortalities investigation at Meatu and Kishapu district, unpublished report submitted to the Officer In-Charge, VIC Mwanza, dated 9 August 2011). The clinical signs that were described by livestock keepers and observed by investigators were swelling of the mandibular area and dewlap, conjunctivitis, lacrimation, nasal discharge, coughing, sneezing, lesions on the lips and nostrils, diarrhoea, and emaciation. The mortality rate was approximately 30%. The farmers reported that sheep were more severely affected than goats, and that animals grazing around Lake Kitangiri were more affected. The disease was new to the farmers, and they associated the outbreak with the presence of scavenging domestic pigs wallowing in the lake. One post mortem examination was carried out on an affected goat, which showed an emaciated and dehydrated carcass with congested lungs, hyperaemic trachea, and bronchi containing froth. Based on the clinical history, clinical signs and post mortem signs, and the cessation of cases after vaccination was carried out, it was concluded that PPRV could be the cause, but serum and tissue samples submitted to the CVL were all negative.

In summary, during 2008, PPR disease was confirmed in Ngorongoro district of northern Tanzania, and serological surveys in the Northern and Lake Zones demonstrated that PPRV had been circulating widely in northern Tanzania. It is likely that PPRV was introduced to northern Tanzania from infected areas of southern Kenya through the movement of live infected animals via trade or cross-border movements of flocks, and the delays in the diagnosis of PPRV and the implementation of control measures led to virus spread across the north of the country [52]. Vaccination campaigns during 2009–2012 reduced the number of outbreaks, but PPRV outbreaks continued to occur on an annual basis due to limited vaccination coverage.

### 3.3. Emergence of PPRV Disease in Southern Tanzania, 2009–2010

PPR disease in sheep and goats was first suspected in southern Tanzania in December 2009, based on interviews conducted with local animal health workers and veterinarians [39]. The first village to be affected was believed to be Likuna village in Newala district, Mtwara region, which borders with Mozambique (Figure 1). The source of infection was suspected to be from 70 goats that were purchased by a livestock trader from the Pugu livestock market, near Dar es Salaam. Three days after arrival, some of these goats died, so to limit losses, the trader sold some of the animals to nearby villages where suspected PPR cases subsequently occurred [39,58]. By March 2010, cases were occurring in the nearby Tandahimba and Masasi districts, probably due to the movement of small ruminants through the local markets, and there was high morbidity and mortality [39,58]. Serological analysis using PPRV H c-ELISA showed that 84% of 180 serum samples collected in Tandahimba district were positive, while only 3% of 125 samples collected in Masasi district were positive [39]. During the same period, another outbreak was also suspected in Namtumbo district in Ruvuma region of southern Tanzania, which neighbours Malawi and Mozambique.

In October 2010, risk-based sero-surveillance was carried out in districts in southern Tanzania bordering Mozambique, Malawi, and Zambia. Out of the 720 sera collected, 36.9% were PPRV antibody positive [39]. Although positive serology in the absence of PPRV vaccination suggested that PPRV was circulating, PPRV was not confirmed as the cause of the disease in southern Tanzania until Muse et al. (2011) investigated clinical cases of suspected PPR in two villages in Tandahimba district in March 2011 [59]. PPRV ribonucleic acid (RNA) was detected in ocular and nasal swabs and tissues from post mortem examination by RT-PCR. Livestock keepers had observed increasing mortality in goats, while sheep were less severely affected. The clinical signs were mainly respiratory with only a few animals affected showing diarrhoea. The main signs observed were fever, lacrimation, nasal discharge, respiratory distress and coughing, and oral and nasal ulceration. Mortality was higher in kids than adults, but all age groups were affected. Nodular skin lesions were observed all over the body, which suggests that the animals were co-infected with sheep and goat pox virus [58].

During March to May 2011, Muse et al. (2012) carried out a serological survey in Newala and Tandahimba districts [58]. In eight purposively selected villages with suspected cases, interviews were conducted with owners of affected goats and sheep, and five randomly selected goats and sheep were blood sampled from each household. Out of 79 households visited, 81.3% in Tandahimba district reported that they had seen suspected PPR cases in their flocks compared to 19.4% in Newala district. A total of 216 serum samples were collected and tested for the PPRV antibody by c-ELISA (type of cELISA not specified). The overall seroprevalence was 31% (95% CI 24.9–37.6%), but a higher proportion of samples collected in Tandahimba were positive (55.5%) than in Newala district (5.7%). Given that no PPRV vaccination had been carried out in southern Tanzania, the high seroprevalence suggested PPRV transmission in the area.

In order to determine whether PPRV had been circulating in southern Tanzania prior to 2009, Mbyuzi et al. (2014) conducted PPRV H c-ELISA on 477 serum samples collected from goats and sheep in 2007 from Lindi and Mtwara regions for RVF surveillance, and 504 sera collected from the affected flocks during the 2009 outbreak investigations in Mtwara region [60]. All the sera collected in 2007 were PPRV antibody negative, providing no evidence of PPRV circulation at that time, while 28.8% of goats and 35.7% of sheep were positive in the 2009 sera samples.

In summary, suspected PPR disease occurred in 2009 in southern Tanzania and was confirmed to be caused by PPRV in 2011. It is likely that the virus was introduced by animal trade from another part of Tanzania via a live animal market in Dar es Salaam and was subsequently spread through small ruminant trade and communal grazing. A delay in the confirmation of the diagnosis led to delays in the implementation of control measures which contributed to the persistence and spread of PPRV [39].

### 3.4. PPRV Disease in Eastern Tanzania, 2010

In April 2010, there was high mortality of sheep and goats from an unknown disease in Ulanga district in the southern part of Morogoro region in the Eastern Zone of Tanzania [39]. Investigations indicated that this could have been caused by PPRV, which could have been introduced to the area by the northward movement of migratory pastoralists from Mtwara and Lindi regions. Out of 200 serum samples collected, 76% were positive by PPRV H c-ELISA [39]. In June–July 2010, outbreaks of suspected PPR disease were reported in the Mvomero district in the north of Morogoro region [61]. An investigation was conducted and clinical cases in goats and sheep were found, with signs of nasal discharge, diarrhoea and oral ulcers, while enlarged and congested gastrointestinal and bronchial lymph nodes were observed during post mortem examinations. PPRV was confirmed in tissue samples by RT-PCR. In order to control the outbreaks in the Southern and Eastern Zones, the Directorate of Veterinary Services supported ring vaccination through the provision of PPRV vaccine to Mtwara and Ruvuma regions (252,000 doses), and to Ulanga in the Morogoro region (74,000 doses), which appeared to limit the spread of the disease [52].

In addition, between November 2014 and January 2015 an outbreak investigation was conducted in Morogoro district where mortality and abortions had been reported in goats. Samples were collected from six goats with clinical signs from Morogoro Urban and Mvomero districts [62]. These samples were subjected to molecular analysis for PPRV identification and PPRV was identified by RT-PCR in one goat from each site. The outbreak started a few days after one farm introduced some apparently healthy goats that had been purchased in Mkongeni market in Mvomero district, and neighbouring farms that were in contact in the grazing area were also affected.

### 3.5. PPRV Disease in Central Tanzania, 2014

In the Dodoma region of Central Tanzania, field and laboratory reports held at Dodoma Veterinary Investigation Centre provided evidence of PPRV circulation in this area from 2014. In August 2014, a disease outbreak was reported among small ruminants in Kongwa district, Dodoma Region, in which 19.5% of the affected animals died (Theodata, 2014 Field work report at Central Zone Investigation Centre, unpublished report). The animals were observed to have diarrhoea, profuse nasal discharge, dyspnoea and anorexia. Out of 30 serum samples collected, 53.3% were positive for PPRV by PPRV c-ELISA (the type of c-ELISA was not specified). In the same month, 65 serum samples were collected from Singida Rural District Council, in the Singida region to the west of Dodoma, of which 20% were sero-positive. All the animals that were blood-sampled originated from areas where the last vaccination had been carried out in 2012. In May 2016, there was a disease outbreak in small ruminants in the Bahi district, Dodoma Region, in which animals showed signs of lacrimation, nasal discharge and diarrhoea. Tissue samples, whole blood and nasal swabs were collected and submitted to the Sokoine University of Agriculture for analysis. PPRV RNA was detected in the blood samples, based on information obtained from laboratory results filed at the Central Zone Investigation Centre (ZVC/DOM/Lab/samp/2016/4/19/2—ZVC Dodoma laboratory result file). In addition, 50 serum samples from the same area were submitted to the Tanzania Veterinary Laboratory Agency (TVLA) and 94% were positive for PPRV by c-ELISA (TVLA/CIDB/PPR/001/2016—ZVC Dodoma laboratory result file).

### 3.6. Evidence of PPRV Infection in Camels and Cattle

Serological evidence of PPRV infection in camels was found in a study carried out in northern Tanzania [63]. This study was carried out in eight districts of Arumeru, Longido, Monduli, Mwanga, Same, Hai, Simanjiro and Kilindi in four regions of northern Tanzania. During June and August 2010, a total of 193 serum samples were collected from 14 camel herds and tested for PPRV antibody by H c-ELISA. The overall seroprevalence was 2.6% (5/193) providing evidence of PPRV infection in camels, although no clinical disease had been observed.

In 2011, when investigating the role of cattle in PPRV epidemiology in northern Tanzania, 266 serum samples were collected from cattle living in close proximity to sheep and goats in randomly selected pastoralist households in eight villages in the northern part of Ngorongoro district, where PPR disease had been confirmed in sheep and goats in 2008 [19]. The overall seroprevalence was 17.3% by H c-ELISA and N c-ELISA, and seroprevalence was higher among cattle aged over three years that had been alive during the 2008 outbreak (26.7%) compared to cattle aged less than two years that had been born after 2008 (5.9%). The seroprevalence at the village level ranged from 7% to 48% [19]. The authors concluded that it was likely that there had been cross-species transmission of PPRV from small ruminants to cattle. In addition, Kivaria et al. (2013) report that PPRV antibody was detected in cattle sharing grazing with small ruminants in the Mkuranga district, Pwani region in eastern Tanzania [39]. It should be noted that in both the camel and cattle serological studies, possible cross-reaction of the PPRV c-ELISA with other morbilliviruses that these species could have been exposed to was not explored, such as canine distemper virus.

### 3.7. Evidence of PPRV Infection in Wildlife

Lembo et al. (2013) also carried out PPRV serological analysis of archived serum samples collected from wild species in northern Tanzania, to investigate possible spillover of PPRV from sheep and goats to wildlife [19]. A total of 243 African buffalo, 59 Thomson’s gazelle (*Eudorcas thomsonii*) and 6 Grant’s gazelle (*Nanger granti*) were sampled from the Serengeti National Park (NP) to the west of Ngorongoro district, Ngorongoro Conservation Area (NCA), in the southern part of Ngorongoro district, and from Arusha and Tarangire NPs to the east of Ngorongoro. Samples were also collected from 23 buffalo in Katavi NP, western Tanzania [19]. The samples were collected during wildlife immobilization operations conducted for rinderpest surveillance, research activities and conservation management prior to 2008 and during 2008–2012. The samples were analysed by PPRV H c-ELISA (Biological Diagnostic Supplies Limited [BDSL]) and none were sero-positive, providing no evidence of PPRV infection in these wild ruminant species in these ecosystems during this time period [19].

In June 2014, a study was carried out looking for evidence of a spillover of PPRV from domestic to wild ruminants in NCA [17]. Eleven sampling sites were purposively selected where resident wild ruminants were present in close proximity to domestic small ruminants, sharing grazing and water sources. Clinical cases of PPR disease were confirmed in sheep and goats in the area during the study period by a PPRV rapid diagnostic test (BDSL) and RT-PCR. Serum samples were collected from 46 wild animals. Overall, 63% of the animals sampled were positive by PPRV H c-ELISA (BDSL), and all herds and species had at least one sero-positive animal, except for Thomson’s gazelle for which only one animal was sampled: African buffalo (10 sampled from 2 herds, 50% positive); Grant’s gazelle (30 sampled from 8 herds, 67% positive); Thomson’s gazelle (1 sampled, 0% positive); wildebeest (*Connochaetes taurinus*) (2 sampled from one herd, 50% positive); impala (*Aepyceros melampus*) (3 sampled from one herd, 100% positive). These results provided the first evidence of PPRV infection in wildlife in Tanzania, although no PPR clinical cases have been reported so far in wild species in East Africa except some unverified reports in free ranging wild species from Sudan [64]. The apparently high seroprevalence in wildlife in close proximity to sheep and goats compared to sero-negative wild animals within national parks [19] supports the hypothesis of a spillover of infection from domestic to wild animals [17]. The possible cross-reaction of the PPRV c-ELISA with other morbilliviruses that these wild species could have been exposed to was not explored, such as canine distemper virus.

### 3.8. Molecular Biology of PPRV in Tanzania

As described earlier, tissue samples collected from goats with clinical signs of PPR from the Ngorongoro district during 2008 were PPRV RT-PCR positive [39]. A partial nucleoprotein gene sequence was obtained, and phylogenetic analysis showed that it was a lineage III virus, and that it clustered with PPRV detected in East Africa (Ethiopia in 1994, Sudan in 1972) as well as from the United Arab Emirates (1986) and Oman (1983).

In 2012–2013, Kgotlele et al. (2014) confirmed PPRV by RT-PCR in samples collected from goats with suspected PPR clinical signs in three villages of Ngorongoro district (Meshili, Piyaya and Malambo) and two villages of Mvomero district (Kauzeni and Dakawa) in Morogoro region in eastern Tanzania [55]. Phylogenetic analysis of partial nucleoprotein gene sequences indicated that the PPRV obtained from northern and eastern Tanzania clustered together in lineage III and were most closely related to PPRV from Ethiopia (1996). However, the sequence obtained in 2009 from Ngorongoro was not included in the analysis, so the relationship to this earlier PPRV in Tanzania is not shown.

During the outbreak investigations conducted by Namtimba in 2014–2015 in Morogoro district in Morogoro region, phylogenetic analysis of the nucleoprotein gene nucleotide sequences showed that the PPRV involved in the outbreak clustered with lineage III viruses and was closely related to those reported by Kgotlele et al., (2014) in Dakawa and Ngorongoro (Figure 3) [61,62].

Misinzo et al. (2015) carried out partial nucleoprotein gene sequencing on the PPRV positive samples collected by Muse et al. (2012) from the 2011 PPRV outbreak in the Tandahimba district, southern Tanzania [14]. Phylogenetic analysis showed that one sequence clustered in lineage II with the Nigeria 75/1 vaccine strain, and a second sequence was most closely related to a lineage IV virus from Turkey, and clustered with viruses from the Middle East, and from Central, East and North Africa.

During the June 2014 study of PPRV at the livestock–wildlife interface in Ngorongoro district, Mahapatra et al. (2015) obtained PPRV RT-PCR positive samples from PPR cases in domestic sheep. Based on partial nucleoprotein gene sequencing it was determined that this was a lineage II virus that clustered with the lineage II virus obtained from southern Tanzania in 2011 [17].

During the investigations of PPR outbreaks in Ngorongoro district in June 2015, partial nucleoprotein gene sequences were obtained from PPRV real-time RT-PCR (RT-qPCR) positive samples [56]. Phylogenetic analysis showed that the sequences obtained clustered with the lineage III PPRV obtained by Kgotlele et al. (2014) from Ngorongoro in 2013, and these shared a common ancestry with a cluster of viruses from Kenya (2011), Uganda (2012 and 2018), and DRC (2018), and a cluster of viruses from Tanzania, including from Morogoro [55].

Lineage II: Tanzania Esieki4 Sheep 2014 [17], KF 672,746 Tanzania, MM89 Goat 2011 [14].Lineage III: KF939644 Tanzania Ngorongoro goat 2013 [55], KT989870 PPRV/TAN/Melela/2014 and KT989871 PPRV/TAN/Magadu/2014 [62], KF939643 Tanzania Dakawa Goat 2013 [55]. The six sequences marked by an asterisk (PPRV/TAN/Goat 3, 4, 10, 11 and Sheep 14, 15/2015 are GenBank accession numbers: MT181842-47 obtained from Ngorongoro 2015 [56].Lineage IV: KF 672,745 Tanzania, MM88 Goat 2011 [14].

In summary, lineage III PPRVs that share a common ancestry with lineage III viruses from Kenya, Uganda and DRC have been identified in northern and eastern Tanzania between 2008 and 2015 (Figure 4). Lineage IV virus has been detected once from southern Tanzania, and closely related lineage II viruses have been detected in the north and the south. This diversity of viruses indicates that there have been multiple introductions of PPRV into Tanzania. PPRV lineage III viruses predominate in East Africa (Sudan, Ethiopia, Kenya, Uganda, Burundi and DRC), while lineage IV viruses have been detected in East (Uganda, South Sudan, Sudan and Ethiopia), Central and North Africa [65]. The detection of lineage II viruses could be due to the circulation of a virus that is a variant of the Nigeria 75/1 vaccine strain, or it is possible that the results are due to laboratory contamination [65]. It is therefore important to continue to obtain samples for genome sequencing from PPRV outbreaks in all parts of Tanzania to better represent the PPR viruses that are circulating, and to gain insights into how PPRV is spread within the country and across borders.

### 3.9. PPRV Serological Surveys

In 2013, a PPRV serological survey was carried out with the aim of determining the sero-prevalence of PPRV in different ecological zones [66]. A total of 32 districts with large sheep and goat populations that had not previously been surveyed, and where no PPRV vaccination had yet been carried out, were purposively selected, of which three (9.4%) were from the coastal zone, 12 (37.5%) were from the semi-arid zone and 17 (53.1%) were from the plateau zone. The serum samples used for analysis were retrieved from the zonal veterinary laboratory. The retrieved samples and documents did not indicate the specific age and vaccination status of the sampled animals. A total of 2490 serum samples were analysed by H c-ELISA, and the overall sero-prevalence was 20.1%, while the district-level sero-prevalence ranged from 0% to 75%. The PPRV sero-prevalence in districts in the semi-arid and coastal zones was found to be significantly higher compared to those in the plateau ecological zone [66]. The possibility that some of the sampled animals could have been brought in from infected and/or vaccinated areas and therefore were exposed to PPRV prior to arrival cannot be excluded.

Kgotlele et al. (2016) also described a PPRV serological survey that was carried out in 2013 that aimed to determine the sero-prevalence in 11 of the 25 mainland regions of Tanzania, and therefore the distribution of PPRV in Tanzania [61]. Samples were collected from healthy goats and sheep in 103 villages in 35 districts (no further details of the survey design were provided). In 2015, three additional regions were surveyed: Arusha, Manyara and Kilimanjaro, in which 15 villages from 3 districts were sampled. Out of 3838 sera collected (2886 goats, 952 sheep), 26.0% were sero-positive by N c-ELISA and there was no difference in sero-prevalence between the sheep and goats. At the regional level, sero-prevalence varied from 2.6% to 70.0%, with the highest prevalence in Northern, Central and Eastern zones, which could be partly attributed to the vaccination that was carried out in these areas (the vaccination status of the sampled animals was not mentioned). All the regions sampled in the Lake Zone and Southern highlands had low sero-prevalence (<10.0%), while the two regions in the Western Zone, which was previously believed to be free of PPRV, had seroprevalences of 9.0% (Kigoma) and 11.4% (Tabora).

In summary, the serological surveys concluded that the occurrence of PPRV in the country shows ecological zone pre-disposition, with sero-prevalence being higher in semi-arid and coastal zones characterised by low relative humidity. They confirmed the presence of antibodies against PPRV in sheep and goats in regions of Tanzania that previously had little to no data on the disease, which indicated that PPRV had spread within Tanzania with the possibility of spread across the border to neighbouring countries.

### 3.10. PPRV Co-Infections with Other Pathogens

Several studies of PPR outbreaks have described co-infections of PPRV and other pathogens. During 2015, Jones et al. (2020) investigated a series of reports of a PPR-like disease in pastoralist small ruminant flocks in Ngorongoro district [56]. Out of 33 outbreak investigations, ten flocks were found to be PPRV positive by either a PPRV rapid diagnostic test (PPRV-RDT, BDSL Irvine Ltd., Irvine, UK) and/or RT-qPCR, and two of these flocks were also positive for bluetongue virus (BTV) by RT-qPCR. Based on clinical signs (but not laboratory confirmation), cases of goat pox and contagious caprine pleuropneumonia (CCPP) were also observed in the area during the study. The confirmed PPRV outbreaks showed a diversity of clinical signs—some flocks primarily showed a respiratory syndrome, while others had a diarrhoea syndrome or a more classical PPR syndrome with nasal discharge, coughing and diarrhoea. In line with the clinical syndrome that they observed, the livestock keepers used different local names for the confirmed cases—local terms for respiratory disease, diarrhoea disease or rinderpest-like disease [56].

Muse et al. (2012) described nodules all over the body as a clinical sign of PPR during the 2011 southern Tanzania PPR outbreak, but a photograph of a clinical case shows the classical cutaneous nodules of goat pox infection [58]. Similarly, Kgotlele et al. (2014) described cutaneous nodules as a clinical sign of PPR disease in Ngorongoro and Morogoro, and provided a photograph of typical cutaneous nodules of goat pox [61]. All of these outbreaks were confirmed as PPRV by RT-PCR; therefore, it is likely that there was co-infection with Capripox virus. In 2016, an investigation was carried out to determine the aetiology of a respiratory disease outbreak causing high mortality in sheep and goats in Loliondo Division of Ngorongoro district [57], part of the study area covered by Jones et al. (2020). Four of the affected villages were visited, where the most frequent clinical signs were nasal discharge and diarrhoea, in some cases tinged with blood, as well as skin nodules and loss of body condition. Nasal swabs collected from 61 animals were examined by a multi-plex RT–qPCR method to detect PPRV, *Mycoplasma capricolum subspecies capripneumoniae* (Mccp, causative agent of CCPP), *Pasteurella multocida*, and Capripoxvirus (causative agent of sheep and goat pox). Out of the 61 samples, 38 (62%) were positive for one or more of the four pathogens, and eight of these (21%) had co-infections. The most frequently detected pathogen was *P. multocida* (33% of animals), while 15% had Capripoxvirus, 13% had PPRV and 13% had Mccp. Out of these, one animal (2%) was co-infected with PPRV and Mccp, one was co-infected with PPRV and Capripox, and one was co-infected with PPRV and *P. multocida*. Four animals (7%) were co-infected with Mccp and Capripox [57]. These studies demonstrate the difficulty that field veterinarians face in making a clinical diagnosis of small ruminant respiratory and diarrhoea syndromes, and the importance of laboratory diagnosis to support differential diagnosis and the identification of mixed infections, which is necessary for determining appropriate interventions, especially in relation to the PPRV eradication programme.

### 3.11. Socio-Economic Impact of PPR Disease

Two studies of the socio-economic impact of PPR disease have been carried in pastoralist and agro-pastoralist production systems of Tanzania. The first was conducted in 2012 in two agropastoralist districts—Tandahimba district in Mtwara Region and Ulanga district in Morogoro Region (Msuya and Kimera, 2013: Peste des petites ruminants (PPR): assessment of socio-economic impacts in the Republic of Tanzania. Working document, No 1., 2013 unpublished report), while the second study was carried out in 2017 in the Ngorongoro district in the Northern Zone which is a pastoralist production system, and the Kibaha and Bagamoyo districts in the Eastern Zone, which have agropastoralist production systems [67]. Based on a survey of households in PPR disease-affected villages, the first study estimated that the average loss per household due to PPR disease was 735,820 Tanzanian shillings (490.6 US$). Extrapolating these losses to the national level, it was estimated that the cumulative losses due to PPR disease between 2009 and 2012 in the eight regions that had been affected was approximately 101.8 billion Tanzanian shillings (US$ 67.9 million), due to small ruminant mortality, milk loss, abortion, premature culling and the cost of disease control. In the second study, data on the losses due to PPR disease were collected using a household survey, which included losses due to mortality, milk loss, abortion and weight loss and cost of control. A spreadsheet model was developed which estimated that the average annual loss per household due to PPR disease in goats was 1,920,924 Tanzanian shillings (approximately 850 US$) and due to PPR disease in sheep, the loss per household was 1,162,562 Tanzanian shillings (approximately 530 US$). These studies suggest that PPR is an economically important disease in Tanzania causing a high economic burden and threatening food security, especially among pastoral communities, because it causes the loss of small ruminants as productive assets, reduces herd size, and causes a loss of potential income. Figure 5 shows the population of sheep and goats per region of Tanzania. There is considerable variation in the numbers of small ruminants by region, and therefore in the potential socio-economic impact of the PPR disease. No benefit-cost analysis of PPRV eradication in Tanzania has been carried out, but based on a global benefit-cost analysis [68], it is likely that the eradication of PPRV from Tanzania would be economically beneficial compared to allowing the disease to continue to be endemic, due to the high losses caused by mortality and morbidity and the cost of on-going control measures. Reduced losses due to the PPR disease would support wealth creation and poverty alleviation, justifying investment in regional and global PPRV eradication.

### 3.12. Challenges for the Control and Eradication of PPRV in Tanzania

This review of the literature related to the introduction and spread of PPRV in Tanzania has highlighted some key challenges that will need to be taken into account when planning interventions for the control and eradication of PPRV in Tanzania. Many of these challenges are also relevant for other emerging and endemic infectious diseases of livestock in the country.

#### 3.12.1. Timely Diagnosis and Intervention

For transboundary animal disease control, early detection and rapid response is fundamental to the control of a newly introduced pathogen. In the case of the introduction of PPRV to Tanzania, this review has shown that there were considerable delays between the receiving of the initial reports of high mortality disease to conducting outbreak investigations and obtaining laboratory results to confirm the diagnosis, and finally, the implementation of the disease control interventions. As a result, in each area of Tanzania where PPRV emerged, infection spread widely before any control measures were carried out, and it is likely that PPRV is now endemic in many parts of Tanzania.

#### 3.12.2. Diagnosis of PPR Disease

PPR disease shows a range of clinical signs that are similar to the signs of other diseases, such as contagious caprine pleuropneumonia, pasteurellosis, bluetongue, and foot-and-mouth disease. It was therefore not surprising that the livestock keepers and veterinarians had difficulty in differentiating the newly introduced PPR disease from other small ruminant diseases that were already present in their areas. The confusion with other diseases, as well as co-infection of PPRV with other pathogens, made clinical diagnosis very difficult. Considering the fact that most disease diagnoses and reports are made by field officers using clinical signs, there is a need for capacity building of field personnel on PPRV clinical diagnosis and differential diagnosis, and this should be supported by the deployment of PPRV rapid test kits and sample collection equipment for laboratory diagnosis and differential diagnosis. In addition, the regional and national laboratories require the skills, equipment, and resources to carry out PPRV diagnosis and differential diagnosis to provide timely results for decision-making.

#### 3.12.3. Extensive Production System with Mobility of Small Ruminants 

Most small ruminant production in Tanzania is under extensive management, either small-holder, pastoral or agro-pastoral production, which involves the movement of animals over shorter or longer distances on a daily or seasonal basis to access pasture, water and markets, and to avoid disease [69]. These movements lead to frequent contact between flocks, within and between districts, regions and countries, which facilitates the transmission of infectious pathogens such as PPRV and leads to the spread and maintenance of disease in Tanzania [53,63]. Periods of drought can cause long distance movements of livestock to access water and grazing, and can increase the interaction between livestock and wildlife at grazing and watering points [53,70].

In the past few decades, pastoralist and agro-pastoralist production systems in Tanzania have been facing a shortage of natural pastures and water for livestock, which has been attributed to climate change and an increase in the human and livestock populations [71]. This has forced agro-pastoral and pastoral communities to migrate into different regions of Tanzania to search for pasture and water, which may have facilitated the spread of transboundary animal diseases to different parts of the country. An example of this has been the migration of pastoralists and agro-pastoralists from the Northern and Lake Zones to more humid areas in eastern and southern parts of Tanzania [69] and an increase in the livestock populations in southern Tanzania, an area that is not a traditional livestock production area [71]. A major influx of pastoralists and agro-pastoralists into the Kilombero and Usangu valleys was linked to a decline in water flow to the Great Ruaha River, which caused an energy crisis because the river is a major source of hydro-electric power generation. This forced the Government to evict the pastoralists and agro-pastoralists from the Usangu and Kilombero valleys in 2007 [72,73]. The evicted pastoralists were ordered to settle in the Lindi region, but because of a quarantine imposed in 2008 due to an RVF outbreak, some of them decided to settle in the Ruvuma and Coastal regions [71]. The movement of evicted pastoralists could have contributed to the southward spread of PPRV to the Mtwara and Ruvuma regions.

#### 3.12.4. Transboundary Small Ruminant Movements

Tanzania has extensive land borders with eight countries in East, Central and Southern Africa, and there is unofficial movement of small ruminants across these borders for the purposes of accessing grazing or water sources, trade, or social reasons. The initial introduction of PPRV into Ngorongoro district has been attributed to the movement of pastoralist animals or the live animal trade with Kenya [39,50,52]. The detection of lineage III PPR viruses that share a common ancestry with the viruses detected in Kenya and Uganda supports this hypothesis, although the small number of PPRV sequences from East Africa is a major limitation for phylogenetic analysis. In addition to lineage III PPRV, the detection of lineage II PPRV in the north and south of Tanzania and lineage IV in the south demonstrates that there have been multiple introductions of PPRV to Tanzania. Lineage II is predominantly found in West and Central Africa and was identified in Uganda in 2007, while lineage IV is the predominant lineage in Asia and has been found in North, East and Central Africa [17].

#### 3.12.5. Small Ruminant Trade

The movement of small ruminants for the live animal trade is a common practice in Tanzania. Livestock keepers or traders bring animals on foot from the villages to primary markets, where animals are either slaughtered, bought by livestock keepers for breeding or fattening, or bought by traders for onward transportation to secondary and tertiary markets, which may involve unofficial movement across international borders. The trade network provides opportunities for disease transmission between flocks in different geographical locations. The live animal trade was mentioned as a possible source of introduction and spread of PPRV in Tanzania [50]. In particular, goats that were bought in a market near Dar es Salaam and transported to southern Tanzania were implicated as the cause of the PPR disease outbreak in the southern region [39,58]. Therefore, it is important for PPRV control to understand the small ruminant trade network patterns, and how the network links small ruminant populations in different parts of Tanzania.

#### 3.12.6. Wildlife

So far, there has been no confirmation of clinical PPR disease or virus maintenance in wildlife species in Tanzania but there is clear evidence of regular spillover from infected livestock. Evidence of clinical disease and spread of PPRV amongst free-ranging wildlife species in Sudan (Dorcas gazelle—*Gazella dorcas*—which is in the same genus as Thomson’s and Grant’s gazelle) [64] and Asia (including impala and Thomson’s gazelle) and in captivity, suggest that wildlife in the Tanzanian ecosystems are for some reason resistant to disease. This may reflect optimal environmental, ecological, and nutritional conditions for wildlife population and the overall robust health of these animals. Whilst this situation persists there is unlikely to be a risk of vectoring or bridging virus between wildlife and livestock, but if the situation changes through environmental or climatic perturbation this status may change. This has been seen with other morbillivirus infections in wildlife such as the canine distemper virus and lions in the Serengeti, where infection occurred sporadically year on year but the disease was only expressed under certain environmental conditions [74]. The continued monitoring of wildlife for disease and evidence of infection will be important during the period of control or elimination of virus under the global eradication campaign for PPRV.

#### 3.12.7. Strengthening of Veterinary Services

The OIE Performance of Veterinary Services (PVS) reports of 2009 and 2017 highlighted that the department of veterinary services in Tanzania is faced with a number of challenges that reduce its efficiency, from financial resources to the organizational structure [75,76]. This limits the capacity for the early detection, diagnosis, and rapid response to contain a transboundary disease such as PPR. The failure of the Veterinary Services to meet the OIE, FAO and World Trade Organization standards in terms of early reporting, identification (diagnosis), surveillance activities, livestock movement, border control, and timely management of disease outbreaks could have contributed to the initial spread of PPRV and the ongoing spread and maintenance of PPRV in different areas of Tanzania. Therefore, there is a need for a further strengthening of the veterinary services chain of command, capacity building of laboratories for efficient and timely disease diagnosis, and a strengthening of cross-border surveillance activities.

## 4. Conclusions

In Tanzania, PPRV was first confirmed and reported to the OIE in December 2008. Serological studies and reports indicate that PPRV may have entered the north of the country periodically before 2008 but this may be confounded by the presence of rinderpest virus up until 1999 or later in the border areas with Kenya and serum cross-reactivity for both viruses. It is likely that PPRV was introduced from neighbouring countries and then spread from the north to the south in Tanzania via the live animal trade and pastoralist movements. Although PPRV antibodies have been detected in sheep and goats in all zones, the highest prevalence of disease is in the Northern Zone where there is a large population of sheep and goats. PPRV antibodies have been detected in cattle, camels, and wildlife, and although no clinical cases have been reported in any of these species, there is an ongoing risk of spillover from sheep and goats at the wildlife–livestock interface and the risk of wildlife disease outbreaks is not negligible. The review shows that PPRV is now endemic in Tanzania, which is causing persistent economic losses in the livestock sector, disturbing livelihoods, and posing a potential threat to biodiversity conservation and the wildlife economy. This justifies investment in a rapid progression to the elimination of the virus from Tanzania in coordination with other countries in the region.

## Figures and Tables

**Figure 1 animals-11-01698-f001:**
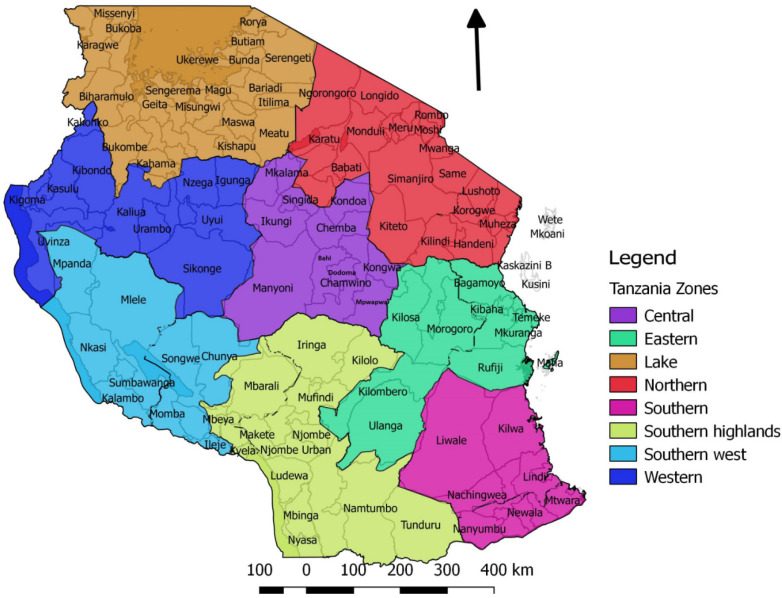
Map of Tanzania showing the seven surveillance zones and their respective districts. Each zone has a Veterinary Investigation Centre and a laboratory.

**Figure 2 animals-11-01698-f002:**
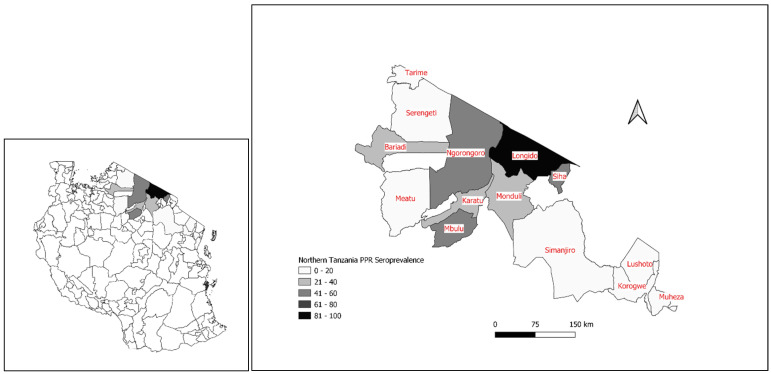
Results of cross-sectional PPRV serological surveys of districts in the Northern and Lake Zones of Tanzania carried out in 2008–2009: map of Tanzania with surveyed districts shaded in grey (**left**); map of surveyed districts indicating district-level seroprevalence (**right**). Source of data: [20,39].

**Figure 3 animals-11-01698-f003:**
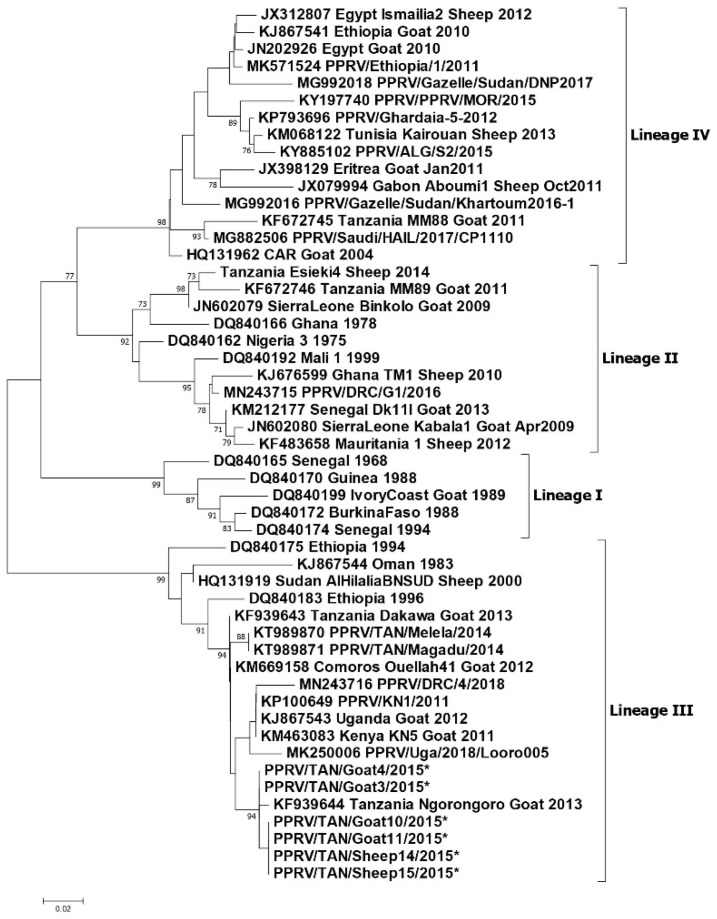
Phylogenetic tree reproduced from Jones et al. (2020) [56]. Neighbour-joining tree constructed on the basis of partial nucleoprotein gene sequences of the peste des petits ruminants virus (PPRV). The tree shows the relationships among African PPRV isolates. The scale bar indicates the nucleotide substitutions per site. The Kimura 2-parameter model with the percentage of replicate trees in which the associated taxa clustered together in the 1000 bootstrap replicates is shown next to the branches. The taxon name of the sequences retrieved from the GenBank contains the accession number followed by the name of the country and the year of isolation. The Tanzanian sequences appearing in this diagram and described in this paper are.

**Figure 4 animals-11-01698-f004:**
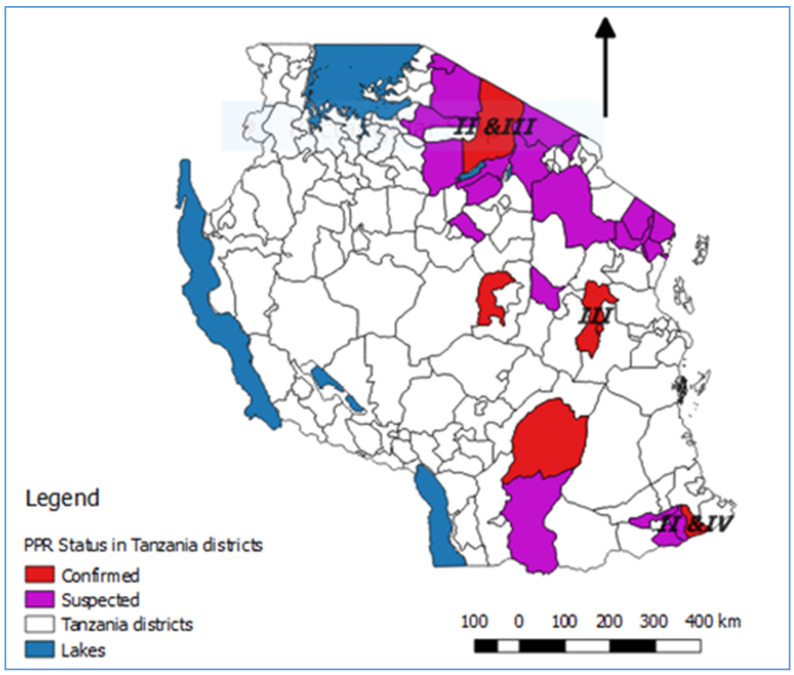
Map of Tanzania showing the districts where PPRV disease was laboratory confirmed or suspected during the period 2008 to 2018. The places where PPRV partial genome sequences were obtained are indicated by the relevant lineage numbers.

**Figure 5 animals-11-01698-f005:**
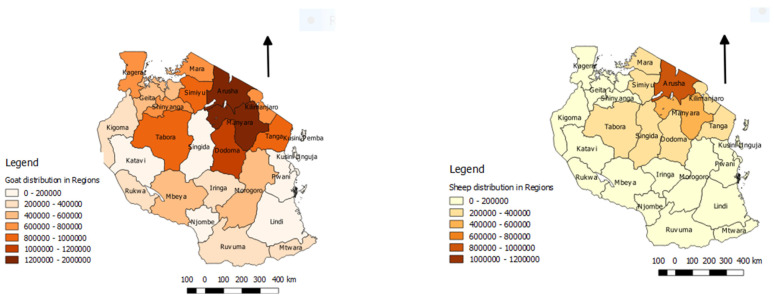
Estimated goat (**left**) and sheep (**right**) populations in different regions of Tanzania. Source of data: Ministry of Livestock and Fisheries, 2020.

## Data Availability

Unpublished reports that are cited in this review that are not publicly available, are available on request from the corresponding author.

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
