# Peer review of "Review of Peste des Petits Ruminants Occurrence and Spread in Tanzania"

_animals, 2021, doi:10.3390/ani11061698_

Round 1

Reviewer 1 Report

This review is very extensive and difficult to follow in some parts. I would suggest you to shorten this manuscript so as to provide only the necessary details in order to achieve the objectives you have set. In addition, it would be very useful if you highlighted some new findings or observations you believe that would be useful for the management of emerging diseases in general.

Author Response

Response to Reviewer 1 Comments

Point 1: This review is very extensive and difficult to follow in some parts. I would suggest you to shorten this manuscript so as to provide only the necessary details in order to achieve the objectives you have set.

Response 1: The review aimed to describe all the available published and unpublished information about PPR occurrence and spread in Tanzania to present a comprehensive objective narrative of the current knowledge, together with a discussion of the implications for national PPR control. To guide the reader, an overview of the themes covered is provided at lines 127 – 133.

Point 2: It would be very useful if you highlighted some new findings or observations you believe that would be useful for the management of emerging diseases in general.

Response 1: Many of the challenges for PPRV control and eradication in Tanzania are also relevant for other infectious livestock diseases, whether emerging or endemic. We have added a comment to this effect at lines 567-571.

Reviewer 2 Report

The aim of the article was to review was to describe the introduction of PPR to Tanzania and its subsequent persistence and spread to different parts of the country. It shows that PPRV is endemic and it is causing persistent economic losses in the livestock sector. Authors made a detailed description of the studies carried out to investigate the disease in the country, from different point of views, which could contribute to more effective planning and implementation of the control and eradication programmes. However, some parts seems repetions of information.

In general the article is well writen and easy to follow.

My only comment is regarding the literature search. Do the authors follow a structured systematic or narrative review, or another type? Authors indicated that they used the search terms “peste des petits ruminants” and “Tanzania”, which was the initial number of articles? When was an article considered relevant (e.g. has the title or the abstract included the search terms)? Was there any limitation (e.g. time period, language) in the searching terms? Were all the authors involved in the selection of articles?

Author Response

Response to Reviewer 2 Comments  

Point 1: Do the authors follow a structured systematic or narrative review, or another type?

Response 1: The manuscript followed principles of narrative reviews, this manuscript captured the history of disease from occurrence and the way it spread into different parts of the country as narrated from introduction to the current status as shown in line 106.

Point 2: Authors indicated that they used the search terms “peste des petits ruminants” and “Tanzania”, which was the initial number of articles?

Response 2: Search terms used were  “Peste des petits ruminants”, and “Tanzania” –Total of 29 published articles and 19  unpublished documents were used as reference of the PPR in Tanzania review,  now provided as supplementary information.

Point 3: When an article was considered relevant (e.g. has the title or the abstract included the search terms)?

Response 3: The article was considered relevant after considering the tittle and later the abstract as indicated in line 113 - 115.

Point 4: Was there any limitation (e.g. time period, language) in the searching terms?

Response 4: For the published articles there were limitation on language (English), papers published in all years up to 2020 were included as shown on line 107. For the grey literature there was no limitation on language, as shown on line 112 to 113.

Point 5: Were all the authors involved in the selection of articles?

Response 5: Among the authors Mdetele, Jones, Komba and Kock involved in selection of articles where the remaining authors involved in providing additional literature. This information has been added in the document see line 117 to 118.

Reviewer 3 Report

The Authors present an interesting review of Peste des petits ruminants (PPR) and the causative agent PPR virus (PPRV) from its first official detection in Tanzania and subsequent spread through the various regions of the country. They also review evidence that suggests the virus was present in Tanzania before it was first officially declared in 2008.

Line 42 Suggest adding keywords; “Peste des petits ruminants” and “Peste des petits ruminants virus” - these terms are the major themes of the review. Also suggest deletion of “PPR” a Google search with this term yields a huge number of results,

Consider deleting “eradication” – while this is mentioned in the review, it is proposed as future goal and therefore arguably not a relevant keyword.

Line 79 The authors state “

Line 99 I would ask the authors to consider providing a flow chart of the decision-making processes and the results thereof to illustrate the number of documents included in the review were compiled. This would not necessarily need to be included in the manuscript e.g., it could be a supplemental file. The figure could also demonstrate how the results of the searches were compiled to the final list of documents for review.

Line 228 Are these really “epidemiological zones”? Or are they geographical, governmental zones etc? Perhaps each zone has a responsible veterinary disease centre?

Line 129 suggest revision “hemagglutinin (H) competitive enzyme-linked immunosorbent assay (c-ELISA)”

Does this assay allow for differentiation of rinderpest virus and PPRV?

The authors should consider adding a few sentences where rinderpest virus is first mention explaining the difficulties in differentiating between the two viruses.

I would ask the authors to carefully consider the terminology used in the manuscript. So there are two diseases (rinderpest and Peste des petits ruminants) and two viruses (rinderpest virus and Peste des petits ruminants virus). Noting other viruses are also mentioned but these are the key ones. The terms of the disease and the virus are used somewhat interchangeably in the manuscript. So the authors might say “antibodies to PPR”, whereas I think it would be more correct to say “antibodies to PPRV” in most instances.

Line 139 Suggest revision: These species are listed in the next sentence with the species names in parenthesis, if the authors wish to use both the common name and scientific name, then this should be done when first described.

Line 144 suggest revision “nucleoprotein (N) c-ELISA”

Line 145 Does this text refer to two “H ELISA” tests? Some of the previous text suggests a single cross-reactive ELISA. If there a specific H ELISA tests for each virus, then terms should be used to clearly indicate this, e.g. for H-PPRV ELISA and H-rinderpest ELISA or something similar.

Line 171. There is nothing on the map “coloured” white – Tanzania districts. Ideally, units should be provided for the scale.

Line 175 There does not appear to a district on the map provide in Figure 1 named “Ngorongoro”.

There is a large unlabelled area in the Northern Zone which is unlabelled which I think corresponds to the Ngorongoro district. I think it is important that given one of the aims of this review is to track the spread of PPRV through Tanzania that any named geographical regions are shown.

Line 179 suggest revision “be antibody negative to PPRV”

Line 248-249 – while accessing grey literature is described in the materials and methods. How accessible are the reports such as Qwari (2011)? Would the authors be able to provide a statement in the manuscript that reports such as this and other grey literature included in the document are available upon request to the corresponding author?

While the materials and methods suggest it is publicly available it is not clear how someone might access these primary resources if they wished to.

Line 325 Similar to my previous comment, the Dooma region nor the Kongwa district do not appear to be shown in the provided maps. Perhaps they are not meant to be, but when one of the objectives of the study is to trace the spread of PPR and PPRV across the country it is useful to be able to view the geography of this. Thus I would encourage the authors to refer to relevant maps in the figures to allow the reader to follow these concepts.

Line 380 The abbreviation for the nucleoprotein should be introduced when the protein is first mentioned (see Line 144). I would suggest not abbreviating, as I am not a fan of abbreviating single-word terms. Plus, it would prevent confusion with the common neuraminidase (N) of other viruses.

Lines 488-502 Is it possible to provide some context for the potential losses attributed to PPR and PPRV? For example, is there an estimate of the total household incomes to provide a perspective for the estimated losses? Similarly, with the total region losses. These figures seem to suggest the disease and virus have an extraordinary impact on these households and economies.

Line 567 – Please provide a reference for the statement regarding the movement of animals from Kenya

Author Response

Response to Reviewer 3 Comments

Point 1: Line 42 Suggest adding keywords; “Peste des petits ruminants” and “Peste des petits ruminants virus” - these terms are the major themes of the review. Also suggest deletion of “PPR” a Google search with this term yields a huge number of results, Consider deleting “eradication” – while this is mentioned in the review, it is proposed as future goal and therefore arguably not a relevant keyword.

Response 1: The words “Peste des petits ruminants” and “Peste des petits ruminants virus  added as well the word eradication has been removed as suggested (see lines 43 – 44).

Point 2: Line 79 The authors state “

Response 2: This meaning of this comment is not clear.

Point 3: Line 99 I would ask the authors to consider providing a flow chart of the decision-making processes and the results thereof to illustrate the number of documents included in the review were compiled. This would not necessarily need to be included in the manuscript e.g., it could be a supplemental file. The figure could also demonstrate how the results of the searches were compiled to the final list of documents for review.

Response 3: This literature review was not a systematic review or meta-analysis, although the search for literature aimed to be as comprehensive as possible. Relevant papers that contributed to the objectives of the literature review were read, relevant information related to PPRV in Tanzania were extracted and compiled into a narrative description of the information. A list of the papers and documents included in the review is now provided as supplementary information.

Point 4: Line 228 Are these really “epidemiological zones”? Or are they geographical, governmental zones etc? Perhaps each zone has a responsible veterinary disease centre?

Response 4: We agree and thank the reviewer for the observation. Tanzania is divided into surveillance zones for animal disease surveillance, each with a zonal veterinary investigation centre and zonal veterinary laboratory. This has been corrected to  “surveillance zones” (see line 138).

Point 5: Line 129 suggests revision “hemagglutinin (H) competitive enzyme-linked immunosorbent assay (c-ELISA)”Does this assay allow for differentiation of rinderpest virus and PPRV?

Response 5: The samples were analysed by both rinderpest virus c-ELISA and PPRV c-ELISA, to differentiate between antibodies against rinderpest virus and those against peste des petits ruminants virus. This has been clarified at lines 139-140.

Point 6: The authors should consider adding a few sentences where rinderpest virus is first mention explaining the difficulties in differentiating between the two viruses.

Response 6: This has now been explained at lines 139-141.

Point 7: I would ask the authors to carefully consider the terminology used in the manuscript. So there are two diseases (rinderpest and Peste des petits ruminants) and two viruses (rinderpest virus and Peste des petits ruminants virus). Noting other viruses are also mentioned but these are the key ones. The terms of the disease and the virus are used somewhat interchangeably in the manuscript. So the authors might say “antibodies to PPR”, whereas I think it would be more correct to say “antibodies to PPRV” in most instances.

Response 7: Regarding the terminology “rinderpest and Peste des petit ruminants” diseases and the two viruses “Rinderpest virus and Peste des petits ruminant  virus”, this has been corrected throughout the document.

Point 8: Line 139 Suggest revision: These species are listed in the next sentence with the species names in parenthesis, if the authors wish to use both the common name and scientific name, then this should be done when first described.

Response 8: This has been corrected (see lines 157-158).

Point 9: Line 144 &145 suggest revision “nucleoprotein (N) c-ELISA”Line 145 Does this text refer to two “H ELISA” tests? Some of the previous text suggests a single cross-reactive ELISA. If there a specific H ELISA tests for each virus, then terms should be used to clearly indicate this, e.g. for H-PPRV ELISA and H-rinderpest ELISA or something similar.

Response 9: The two tests H c-ELISA and the N c-ELISA tests are two different ELISAs where the respective H c-ELISA can differentiate the rinderpest virus antibodies from PPRV antibodies. The sentence has been changed in the document to show the difference and existence of specific H c-ELISA for RPV and PPRV (see lines139-145).

Point 10: Line 171. There is nothing on the map “coloured” white – Tanzania districts. Ideally, units should be provided for the scale.

Response 10: The improved map is inserted on the document with units on the scale and word respective district added to the title of the map (see Figure 1)

Point 11: Line 175 There does not appear to a district on the map provide in Figure 1 named “Ngorongoro”. There is a large unlabelled area in the Northern Zone which is unlabelled which I think corresponds to the Ngorongoro district. I think it is important that given one of the aims of this review is to track the spread of PPRV through Tanzania that any named geographical regions are shown.

Response 11: The improved map is inserted on the document with Ngorongoro district labelled (see Figure 1).

Point 12: Line 179 suggest revision “be antibody negative to PPRV”

Response 12: The sentence has been revised in the document it reads “antibody negative to PPRV” (see line 197).

Point 13: Line 248-249 – while accessing grey literature is described in the materials and methods. How accessible are the reports such as Qwari (2011)? Would the authors be able to provide a statement in the manuscript that reports such as this and other grey literature included in the document are available upon request to the corresponding author?While the materials and methods suggest it is publicly available it is not clear how someone might access these primary resources if they wished to.

Response 13: A data availability statement has been added (lines 670-671) indicating that unpublished reports that are not publicly available, are available on request from the corresponding author.

Point 14: Line 325 Similar to my previous comment, the Dooma region nor the Kongwa district do not appear to be shown in the provided maps. Perhaps they are not meant to be, but when one of the objectives of the study is to trace the spread of PPR and PPRV across the country it is useful to be able to view the geography of this. Thus I would encourage the authors to refer to relevant maps in the figures to allow the reader to follow these concepts.

Response 14: The improved map has been inserted showing the districts which were missing including Kongwa, Bahi and Dodoma (see Figure 1).

Point 15: Line 380 The abbreviation for the nucleoprotein should be introduced when the protein is first mentioned (see Line 144). I would suggest not abbreviating, as I am not a fan of abbreviating single-word terms. Plus, it would prevent confusion with the common neuraminidase (N) of other viruses.

Response 15: As suggested, we have used the full term “nucleoprotein” when referring to nucleoprotein gene sequences – this has been corrected throughout this section.

Point 16: Lines 488-502 Is it possible to provide some context for the potential losses attributed to PPR and PPRV? For example, is there an estimate of the total household incomes to provide a perspective for the estimated losses? Similarly, with the total region losses. These figures seem to suggest the disease and virus have an extraordinary impact on these households and economies.

Response 16: Unfortunate the two referred documents has not shown the estimate of the total household incomes and region losses.

Point 17: Line 567 – Please provide a reference for the statement regarding the movement of animals from Kenya

Response 17: The references regarding movement of small ruminant between Kenya and Tanzania have been added (see line 616).

Reviewer 4 Report

Specific Comments

Line 47: Identifies Small ruminant morbillivirus as the cause of PPR, However, Line 49 and numerous subsequent lines refer to PPR virus (PPRV). This needs to be reconciled for consistency.

Line 62 -63 reads “The virus also causes sub-clinical infection in cattle with the development of a cross-neutralizing and cross-protective humoral response against PPR”. A perusal of the references (18 and 19) cited against this sentence does not support this assertion. The serum samples collected in both studies were tested using the anti-haemaglutinin (H) c-ELISA test. No virus neutralization tests were carried out in either of the cited studies and therefore the reference to a cross-neutralizing and cross protective humoral response against PPR is not supported by these two references contrary to what is implied by the authors. Furthermore, infection with small ruminant morbillivirus (formerly PPRV) in cattle cannot cross-neutralize and cross-protect against PPR. This would be relevant if the reference was to another closely related agent or disease other than PPR.

Line 70: Vaccination with live attenuated PPR vaccines also provides lifelong protection (it is not all vaccinations that confer lifelong protection).

Line 128-129: Gives the impression that one H c-ELISA test was used for the detection of rinderpest and PPR antibodies. There are two distinct H c-ELISA tests for rinderpest and PPR. I therefore suggest that the authors add the word respective before “H competitive enzyme-linked immunosorbent assay”.

Line 135: Explain what is “N c-ELISA” at the first mention.

Line 204: Explain partial N gene sequencing (what is the N gene). Not all readers will be familiar with the molecular structure of Small ruminant morbillivirus (PPRV).

Line 230: to clarify which c-ELISA (H or N) was used for analysis and also for subsequent references to c-ELISA tests in the manuscript that are not supported by the citation of a published reference.

Line 240: To indicate under which institution the VACNADA project was anchored as indicated under line 236 (VETAID) and Line 239 (FAO).

Lines 282 – 283: Indicate in full what are “RNA and RT-PCR” at the first mention.

Lines 339 – 353 and 354 - 377: Were any studies done to rule out the possibility of cross-reactions with other morbilliviruses in the cattle, camels and wildlife samples tested?

Lines 389 – 393 could all be merged into one sentence to avoid the repetition in the last sentence (391-393) relating to the phylogenetic analysis of the N gene nucleotide sequences.

Lines 438 -457: What was the sampling design for the serological surveys, what age groups of animals were sampled and how did the authors preclude the possibility of sampling previously vaccinated sheep and goats or young animals with maternally derived antibodies to PPR virus, particularly for the surveys carried out in 2013.

Line 461: Indicate in full what are “PPRV-RDT and RT-qPCR” at the first mention.

Line 463: Indicate in full what is “CCPP” at the first mention.

Lines 488 – 492: What production systems did the socio-economic impacts cover and what parameters were assessed to determine losses at the household levels and effects at the national level.

Line 496: Kindly explain what is a morbidity loss?

Lines 587 -588: Were the wildlife species observed in Tanzania the same as those involved in clinical PPR outbreaks in Sudan and Asia? There may also have been differences in species susceptibility?

Lines 616 – 617: The conclusion in relation to the threat of PPR to biodiversity conservation and the wildlife economy is Tanzania is not supported by the evidence presented in this manuscript as no losses have been attributed to PPR in the wildlife populations examined.  

Author Response

Response to Reviewer 4 Comments

Point 1:Line 47: Identifies Small ruminant morbillivirus as the cause of PPR, However, Line 49 and numerous subsequent lines refer to PPR virus (PPRV). This needs to be reconciled for consistency.

Response 1: The virus belongs to Genus Morbillivirus, which is in the family Paramyxoviridae. The commonly used name is peste des petits ruminants virus (PPRV) but the taxonomic name is Small ruminant morbillivirus. This has been clarified (line 49).

Point 2:Line 62 -63 reads “The virus also causes sub-clinical infection in cattle with the development of a cross-neutralizing and cross-protective humoral response against PPR”. A perusal of the references (18 and 19) cited against this sentence does not support this assertion. The serum samples collected in both studies were tested using the anti-haemaglutinin (H) c-ELISA test. No virus neutralization tests were carried out in either of the cited studies and therefore the reference to a cross-neutralizing and cross protective humoral response against PPR is not supported by these two references contrary to what is implied by the authors. Furthermore, infection with small ruminant morbillivirus (formerly PPRV) in cattle cannot cross-neutralize and cross-protect against PPR. This would be relevant if the reference was to another closely related agent or disease other than PPR.

Response 2: Thank you for pointing out this mistake. This sentence has now been corrected to state that cattle exposed to PPRV developed antibodies to PPRV (lines 65-66).

Point 3:Line 70: Vaccination with live attenuated PPR vaccines also provides lifelong protection (it is not all vaccinations that confer lifelong protection).

Response 3: This has been clarified at lines 73-74 – “live attenuated PPRV vaccine provides lifelong protection”.

Point 4:Line 128-129: Gives the impression that one H c-ELISA test was used for the detection of rinderpest and PPR antibodies. There are two distinct H c-ELISA tests for rinderpest and PPR. I therefore suggest that the authors add the word respective before “H competitive enzyme-linked immunosorbent assay”.

Response 4: Yes, two distinct assays were used. This has been clarified at lines 139-140 – “The samples were analysed for both rinderpest virus and PPRV antibodies using the rinderpest virus haemagglutinin (H) competitive enzyme-linked immunosorbent assay (c-ELISA) and the PPRV H c-ELISA [47].

Point 5:Line 135: Explain what is “N c-ELISA” at the first mention

Response 5: The sentence, “PPRV antibodies on plates coated with a PPRV recombinant nucleocapsid protein (N) c-ELISA ..”  has been added (see line 152).

Point 6:Line 204: Explain partial N gene sequencing (what is the N gene). Not all readers will be familiar with the molecular structure of Small ruminant morbillivirus (PPRV).

Response 6: To clarify this point, the statement “using primers targeting the nucleoprotein (N) gene, showing that this virus belonged to lineage III” has been added at lines 222-223.

Point 7:Line 230: to clarify which c-ELISA (H or N) was used for analysis and also for subsequent references to c-ELISA tests in the manuscript that are not supported by the citation of a published reference.

Response 7: The type of c-ELISA – N or H – has been specified where this information was available. Not all sources specified which c-ELISA was used. Where there is uncertainty, this has been clarified by adding “(type of c-ELISA not specified)” throughout the manuscript.

Point 8:Line 240: To indicate under which institution the VACNADA project was anchored as indicated under line 236 (VETAID) and Line 239 (FAO).

Response 8: The VACNADA project was coordinated by AU-IBAR and funded by the European Commission. In Tanzania it was implemented by Veterinary services of Tanzania. This information has been added (see lines 259-260).

Point 9:Lines 282 – 283: Indicate in full what are “RNA and RT-PCR” at the first mention.

Response 9: RNA has been written in full on line 303 (ribonucleic acid). RT-PCR is already written in full earlier in the manuscript (line 222).

Point 10:Lines 339 – 353 and 354 - 377: Were any studies done to rule out the possibility of cross-reactions with other morbilliviruses in the cattle, camels and wildlife samples tested?

Response 10: The serological studies in cattle and camels only used PPRV specific c-ELISAs, similarly for the wildlife study (Mahapatra et al, 2015). For the wildlife study of Lembo et al (2013), some sera were collected for rinderpest surveillance but the results of the rinderpest serology are not reported in the paper. This information has been added at lines 377-379, and lines 404-405.

Point 11:Lines 389 – 393 could all be merged into one sentence to avoid the repetition in the last sentence (391-393) relating to the phylogenetic analysis of the N gene nucleotide sequences.

Response 11: As suggested these sentences have been merged – see lines 420-421.

Point 12:Lines 438 -457: What was the sampling design for the serological surveys, what age groups of animals were sampled and how did the authors preclude the possibility of sampling previously vaccinated sheep and goats or young animals with maternally derived antibodies to PPR virus, particularly for the surveys carried out in 2013.

Response 12: For the survey described in Mdetele et al (2020), as described in line 464, the study was carried out using stored serum samples collected from goats and sheep in the year 2013. During implementation of the project, sampling was performed in 32 different districts in which PPR surveillance activities had never been carried out before to assess if goats and sheep on those areas have ever encountered the disease after the first outbreak of the disease in Tanzania in the year 2008. The retrieved samples and sampling document did not indicate age and vaccination status of sampled animals the information which does not rule out the possibility of effect of vaccination and maternal immunity. However, it was assumed that the sampled animals had never encountered the PPR challenge.

For the survey described in Kgotlele et al 2016, no mention is made of the age or vaccination status of the animals sampled, although they acknowledge that vaccination campaigns had been carried out in some of the areas that had higher seroprevalence. This information has been clarified at lines 476-478 and lines 485-487.

Point 13:Line 461: Indicate in full what are “PPRV-RDT and RT-qPCR” at the first mention

Response 13: The acronym PPRV-RDT has now been given in full (PPRV rapid diagnostic test) at lines 498-499. RT-qPCR was given in full earlier (line 433).

Point 14:Line 463: Indicate in full what is “CCPP” at the first mention.

Response 14: The acronym CCPP has been given in full (contagious caprine pleuropneumonia) at line 501.

Point 15:Lines 488 – 492: What production systems did the socio-economic impacts cover and what parameters were assessed to determine losses at the household levels and effects at the national level.

Response 15: The production systems considered were pastoralist and agropastoralist – this information has been added at lines 527-528, and lines 532-533. The parameters considered in Msuyu and Kimera (2013) were losses due to death of sheep and goats, losses due to morbidity (abortion, weight loss, milk loss) and the cost of disease control (treatment, vaccinations and premature culling). The parameters considered by George (2017) were Mortality loss, Abortion, Weight loss, Milk loss and treatment loss. The table showing those losses and amount has been inserted..This information has been added in the manuscript at lines 537-, and lines 539-540.

Point 16: Line 496: Kindly explain what is a morbidity loss?

Response 16: Morbidity loss is the loss due to animals becoming sick (but not dying) – see the details of morbidity losses in response 15 above.

Point 17:Lines 587 -588: Were the wildlife species observed in Tanzania the same as those involved in clinical PPR outbreaks in Sudan and Asia? There may also have been differences in species susceptibility?

Response 1: The wildlife species affected in Sudan was Dorcas gazelle, which are in the same genus as Grant’s and Thomson’s gazelles. In Asia, artiodactyl species that are indigenous to Africa have been infected and suffered from PPR disease, including species found in Tanzania – impala and Thomson’s gazelle. This information has been added at lines 637-638.

Point 18:Lines 616 – 617: The conclusion in relation to the threat of PPR to biodiversity conservation and the wildlife economy is Tanzania is not supported by the evidence presented in this manuscript as no losses have been attributed to PPR in the wildlife populations examined.

Response 18:

Response 18:There is a potential threat of PPR to biodiversity conservation and the wildlife economy because although no disease has so far been detected in wild species in Tanzania, clinical disease has occurred in species found in this ecosystem in the Middle East, and related species have been affected in Sudan. Given adverse environmental conditions or other stressors (as mentioned in lines 639-642),  virus spillover from domestic animals in close proximity with wildlife could lead to disease and death of wildlife. We have added the word “potential” at line 668 to clarify this.

Round 2

Reviewer 1 Report

No comments